# Retinal direction selectivity in the absence of asymmetric starburst amacrine cell responses

**Laura Hanson[†], Santhosh Sethuramanujam[†], Geoff deRosenroll, Varsha Jain, Gautam B Awatramani\***

Department of Biology, University of Victoria, Victoria, Canada

**Abstract** In the mammalian retina, direction-selectivity is thought to originate in the dendrites of GABAergic/cholinergic starburst amacrine cells, where it is first observed. However, here we demonstrate that direction selectivity in downstream ganglion cells remains remarkably unaffected when starburst dendrites are rendered non-directional, using a novel strategy combining a conditional GABA$_A$ $\alpha$2 receptor knockout mouse with optogenetics. We show that temporal asymmetries between excitation/inhibition, arising from the differential connectivity patterns of starburst cholinergic and GABAergic synapses to ganglion cells, form the basis for a parallel mechanism generating direction selectivity. We further demonstrate that these distinct mechanisms work in a coordinated way to refine direction selectivity as the stimulus crosses the ganglion cell's receptive field. Thus, precise spatiotemporal patterns of inhibition and excitation that determine directional responses in ganglion cells are shaped by two 'core' mechanisms, both arising from distinct specializations of the starburst network.

DOI: https://doi.org/10.7554/eLife.42392.001

**\*For correspondence:**
gautam@uvic.ca

[†]These authors contributed equally to this work

**Competing interests:** The authors declare that no competing interests exist.

## Introduction

The direction-selective (DS) circuit in the retina is arguably one of the most well-defined circuits in the mammalian brain (reviewed by *Mauss et al., 2017*; *Vaney et al., 2012*). In this circuit, direction is encoded by output DS ganglion cells (DSGCs), whose response properties are shaped by DS inputs arising from GABAergic/cholinergic starburst amacrine cells (starbursts) and non-DS inputs from glutamatergic bipolar cells (*Figure 1A*). However, despite comprehensive connectomic analysis of the circuit (*Briggman et al., 2011*; *Ding et al., 2016*) and many decades of physiological investigations, the fundamental mechanisms underlying direction selectivity, both at the level of starbursts and DSGCs remain to be fully elucidated.

The first points in the visual system where direction selectivity is observed are the radiating dendrites of starbursts (*Euler et al., 2002*), which release GABA and acetylcholine (ACh) (*O'Malley and Masland, 1989*). To explain direction selectivity, initial studies emphasized mechanisms that were based on intrinsic properties of starburst dendrites (*Gavrikov et al., 2006*; *Hausselt et al., 2007*; *Tukker et al., 2004*). In contrast, more recent studies have proposed two distinct network mechanisms. The first relies on inhibition within the dense array of starburst cells, where dendrites of neighboring starbursts with opposite directional preferences mutually inhibit each other. This ensures that dendrites pointing in a particular direction are maximally activated as the moving stimulus sequentially activates the starbursts (*Ding et al., 2016*; *Lee and Zhou, 2006*; *Münch and Werblin, 2006*). The second network mechanism relies on the specific wiring of temporally distinct bipolar cells along the proximal-distal axis of the starburst dendrite. This arrangement results in an optimal summation of inputs when the stimulus moves centrifugally (from soma-to dendrite) along the starbursts dendrites (*Fransen and Borghuis, 2017*; *Kim et al., 2014*). When tested experimentally, however,

neither of these mechanisms appears to be critically required for generating DS, at least in the context of simple spot/grating stimuli (*Chen et al., 2016*; *Sethuramanujam et al., 2016*).

Starbursts play an obligatory role in generating direction selectivity in downstream DSGCs (*Vlasits et al., 2014*; *Yoshida et al., 2001*). They provide the neural substrate for a powerful GABAergic 'null' inhibition that is required for the generation of DS responses in ganglion cells (*Figure 1C*). Null inhibition relies on both the DS properties of starburst dendrites and also a specific 'anti-parallel' wiring scheme (*Figure 1B*). Only starbursts with somas displaced to the null-side of the DSGC's receptive field (i.e. the side from which null-stimuli enter the receptive field) make synaptic connections with DSGCs (*Briggman et al., 2011*; *Brombas et al., 2017*; *Chen et al., 2016*; *Fried et al., 2002*; *Lee et al., 2010*; *Yonehara et al., 2011*). By contrast, excitatory inputs to DSGCs arising from bipolar cells are non-DS (*Park et al., 2014*; *Yonehara et al., 2013*). Together, these findings have given rise to a well-accepted model in which direction selectivity in DSGCs is largely inherited from starburst dendrites, contingent on the DS release of GABA (*Figure 1C*) (reviewed by *Mauss et al., 2017*; *Vaney et al., 2012*).

However, in addition to GABA, starbursts also co-release ACh (*O'Malley and Masland, 1989*), which mediates fast excitatory postsynaptic currents (EPSCs) in DSGCs via the activation of nicotinic ACh (nACh) receptors (*Brombas et al., 2017*; *Lee et al., 2010*; *Pei et al., 2015*; *Sethuramanujam et al., 2016*; *Taylor and Smith, 2012*). Unlike GABA, however, cholinergic signals appear to be more symmetrical at the level of DSGCs, giving rise to the notion it modulates the amplitude of DSGC responses (*Ariel and Daw, 1982*; *Amthor et al., 1996*; *Chiao and Masland, 2002*; *Park et al., 2014*). Recent voltage-clamp recordings on the other hand, reveal amplitude differences between preferred and null-evoked cholinergic currents (Figure 1C), suggesting that ACh may directly contribute to direction selectivity (*Lee et al., 2010*; *Pei et al., 2015*). However, the mechanisms underlying DS cholinergic excitation remain unknown.

Elegant starburst/DSGC paired recordings demonstrate that cholinergic inputs arise from starbursts surrounding DSGCs from all directions, in sharp contrast to GABA inputs (*Figure 1A*) (*Brombas et al., 2017*; *Chen et al., 2016*; *Lee et al., 2010*; *Yonehara et al., 2011*). Given the asymmetric anatomical wiring of the starbursts and DSGCs (*Figure 1B*), it is presumed that cholinergic excitation is mediated by paracrine mechanisms that are agnostic to the specific connectivity (*Briggman et al., 2011*; *Brombas et al., 2017*). Building on these findings, in this study we hypothesized that the differential functional connectivity of spatially offset GABA and ACh signals produces excitation-inhibition (E/I) timing differences that generate direction selectivity, as originally envisioned (*Koch et al., 1982*; *Taylor et al., 2000*; *Torre and Poggio, 1978*). This could serve as a parallel 'hard-wired' DS mechanisms that does not necessarily require DS starburst output (*Figure 1D*).

It is important to note that temporal E/I asymmetries have often been noted in the literature, but their impact has been difficult to assess, as they are always associated with changes in E/I amplitude ratio (*Figure 1D*) (*Fried et al., 2005*; *Kostadinov and Sanes, 2015*; *Pei et al., 2015*; *Sethuramanujam et al., 2016*; *Taylor and Vaney, 2002*). Further, blocking cholinergic inputs, which is expected to decrease E/I timing differences arising from the above-mentioned specific wiring of starburst ACh/GABA synapses (*Figure 1D*), does not appear to affect the ability of DSGCs to encode direction (*Ariel and Daw, 1982*; *Kittila and Massey, 1997*). It has also been shown that robust direction selectivity is observed in instances where E/I temporal offsets are not apparent (*Taylor and Vaney, 2002*). Additionally, modeling studies suggest that E/I temporal offsets play a negligible role in this computation in the context of the E/I amplitude modulation (*Schachter et al., 2010*). Thus, the contribution of E/I temporal offsets to direction coding by DSGCs are generally neglected.

Here, we describe two major advancements that better our understanding of the DS circuitry in the mouse retina. First, using a novel combination of optogenetics/mouse KO technology/pharmacology, we render starburst output non-DS for the first time. This provides valuable insights into the biophysical mechanisms underlying direction selectivity in starburst dendrites. Second, we found that blocking starburst DS had a surprisingly weak effect on direction selectivity in ganglion cells. This demonstrates the existence of a second 'core' DS mechanism, which we demonstrate relies on E/I temporal offsets. Interestingly, while changes in E/I ratios or timing differences are each sufficient to drive robust DS responses in ganglion cells, these mechanisms contribute to distinct phases of the DSGC response, ensuring that direction is computed rapidly and with high fidelity.

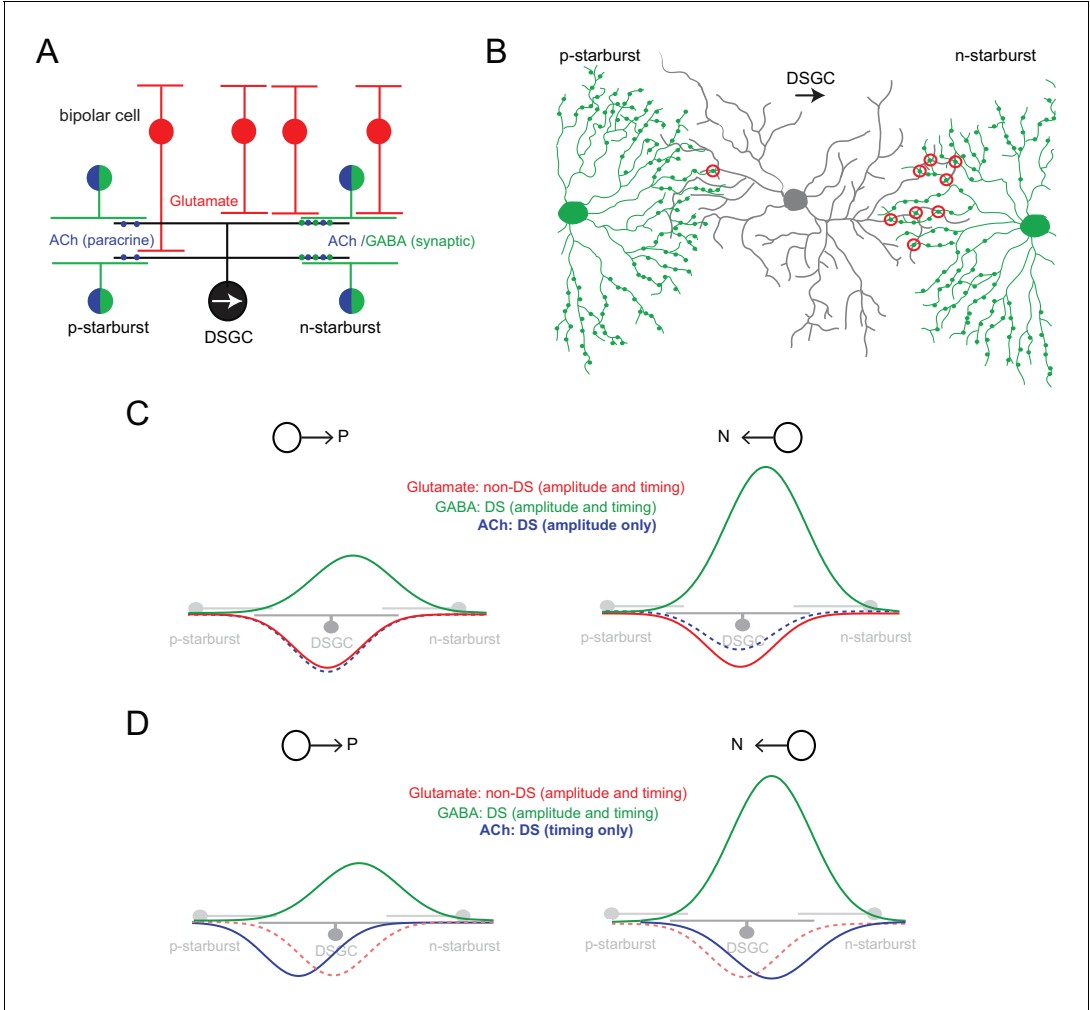

**Figure 1.** Parallel 'core' mechanisms generating direction selectivity in DSGCs (A) A cross section of the functional circuitry underlying direction selectivity in ON-OFF DSGCs. DSGCs receive glutamatergic inputs from ON and OFF bipolar cells, and inputs from mirror symmetric populations of ON and OFF GABAergic/cholinergic starburst amacrine cells. In this study, the stimuli used emphasized the ON responses. Preferred-side starbursts (p-starburst) provide mainly cholinergic excitation, while null-side starbursts (n-starburst) provide cholinergic excitation and a dominant GABAergic inhibition (*Lee et al., 2010*; *Yonehara et al., 2011*). Thus, GABAergic/cholinergic signals mediated by starbursts are differentially transmitted to the DSGC. (B) Schematic depicting the asymmetric anatomical connectivity between starbursts and DSGCs (top view). *Bona fide* 'wrap-around' synaptic connections (circled in red) are made largely by null-side starbursts, enabling DS dendrites of starbursts to mediate a 'null' inhibition (*Briggman et al., 2011*). This contrasts with the symmetrical functional cholinergic connectivity depicted in (A). Paracrine or 'volume' transmission could make cholinergic signals agnostic to the specific synaptic connectivity (*Briggman et al., 2011*; *Brombas et al., 2017*). (C) In conventional models, direction selectivity in DSGCs is largely shaped by asymmetric postsynaptic GABAergic inhibition. Asymmetric inhibition is contingent on the DS release of GABA from dendrites of starbursts; and on the asymmetric wiring (B). By contrast, excitation is non-directional, mainly mediated by glutamate released from bipolar cells (*Park et al., 2014*; *Yonehara et al., 2013*). In this model, the cholinergic receptive field is co-extensive with the glutamatergic receptive field. ACh is usually thought to play a non-directional role (*Ariel and Daw, 1982*; *Amthor et al., 1996*; *Chiao and Masland, 2002*; *Park et al., 2014*), although some studies note amplitude differences between preferred and null-evoked cholinergic currents (as shown here) suggesting that ACh may directly contribute to direction selectivity (*Lee et al., 2010*; *Pei et al., 2015*). (D) In the revised model proposed here, cholinergic excitation is directional by virtue of its timing with GABAergic inhibition rather than by its response amplitude (C). The differences in the functional connectivity of GABA and ACh (A) predict that for preferred-direction motion, excitation would lead inhibition; and for null-direction motion, E/I would be activated together. In this study, the central hypothesis is that E/I offsets contribute to a parallel DS mechanism, which does not necessarily rely on the modulation of the peak amplitude of GABA or ACh inputs (in contrast to the model shown in C).

DOI: https://doi.org/10.7554/eLife.42392.002

## Results

### Rendering starburst dendrites non-directional

We first sought to develop ways to abolish direction selectivity in starburst dendrites to test its contribution to direction selectivity in the postsynaptic DSGCs. Previous attempts to perturb DS in starburst dendrites have been met with little success. For example, selectively eliminating mutual inhibition between anti-parallel starburst dendrites using the GABA$_A$ α2 receptor KO mouse line (*Gabra2* KO) leaves starburst DS largely intact (*Chen et al., 2016*). Consistent with these findings, we found the DS output of the starbursts measured as inhibitory postsynaptic currents (IPSCs) in DSGCs, was similar in *Gabra2* KO mice and wild type mice. This was quantified using a direction selectivity index (DSI; calculated as the normalized vector sum of responses measured across eight directions; Wt = 0.33 ± 0.019, n = 6; KO = 0.28 ± 0.022; n = 6; data are presented as mean ± SEM; *Figure 2A,B,E,F*).

Arguing against excitatory network mechanisms (*Fransen and Borghuis, 2017*; *Kim et al., 2014*), robust DS output was observed in the absence of bipolar cell input, when the starbursts were directly stimulated using optogenetics (*Figure 2C,E & F*; *Sethuramanujam et al., 2016*). In these experiments, photoreceptor/bipolar cell driven responses were blocked using conventional pharmacology (50 μM DL-AP4, 10 μM UBP310 and 20 μM CNQX) and the stimulus intensity was increased 1000-fold (R*/s) to directly activate starburst-expressing channelrhodopsin2 (ChR2) in relative isolation. While there was considerable variability in the amplitude of IPSCs, which may have arisen from variable expression of ChR2 and/or run down of optogenetic responses over the course of the recording session (which could last up to 8 hr), within a given cell the responses and their direction selectivity remained reliable between trials. We thus included weaker responding cells in this analysis.

Remarkably, combining these two approaches to block both inhibitory and excitatory network DS mechanisms, led to the near-complete block of the asymmetry in responses mediated by starburst cells, i.e. the amplitudes of inhibitory inputs evoked by optogenetic stimulation of starbursts in the *Gabra2* KO background were nearly equal for preferred and null-direction motion (DSI = 0.07 ± 0.02, n = 7; *Figure 2D–F*). The ability to block direction selectivity in starbursts, while leaving its output relatively intact, provides for the first time direct evidence for the mechanisms generating it. The requirement to block both the excitatory (*Fransen and Borghuis, 2017*; *Kim et al., 2014*) and inhibitory network mechanisms (*Ding et al., 2016*; *Lee and Zhou, 2006*; *Münch and Werblin, 2006*) suggest that they work in parallel to shape DS responses in starburst dendrites.

### Retinal direction selectivity in the absence of asymmetric starburst amacrine cell responses

Abolishing the directional properties of starbursts is expected to suppress the output of DSGCs, as under these conditions DSGCs would receive strong inhibition in all directions. Contrary to this notion, we found that DSGCs in *Gabra2* KO mice continued to exhibit robust spiking responses when starburst output was rendered non-DS (*Figure 3A*). Strikingly, these spiking responses were robustly tuned for direction similar to control conditions (*Figure 3B–D*). While the direction encoded was the same under conditions in which starburst output was DS or non-DS (*Figure 3A*), we did observe a significant reduction in the duration of the DSGC's spiking response (*Figure 3A*). Since starbursts are critically required for DS computation (*Vlasits et al., 2014*; *Yoshida et al., 2001*), it follows that they must utilize an alternate mechanism to confer direction selectivity upon DSGCs, in the absence of amplitude modulation of inhibitory inputs. In addition, the finding that the direction tuning properties of the DSGC remains unchanged when starburst output is rendered non-DS (*Figure 3B,D*), indicates that this second DS mechanism is well-aligned to the classical mechanism relying on the direction selectivity of starburst dendrites.

We hypothesized that asymmetries in the timing of excitation and inhibition arising from the differential functional starburst-DSGC connectivity gives rise to a parallel DS mechanism (*Figure 1D*). When we examined the onset latencies for GABA and ACh in the *Gabra2* KO, we found E/I temporal offsets were exquisitely tuned for direction (*Figure 4A–D*). The magnitude of the offsets provided a good indication of the DSGC's preferred-direction when compared to its spiking responses

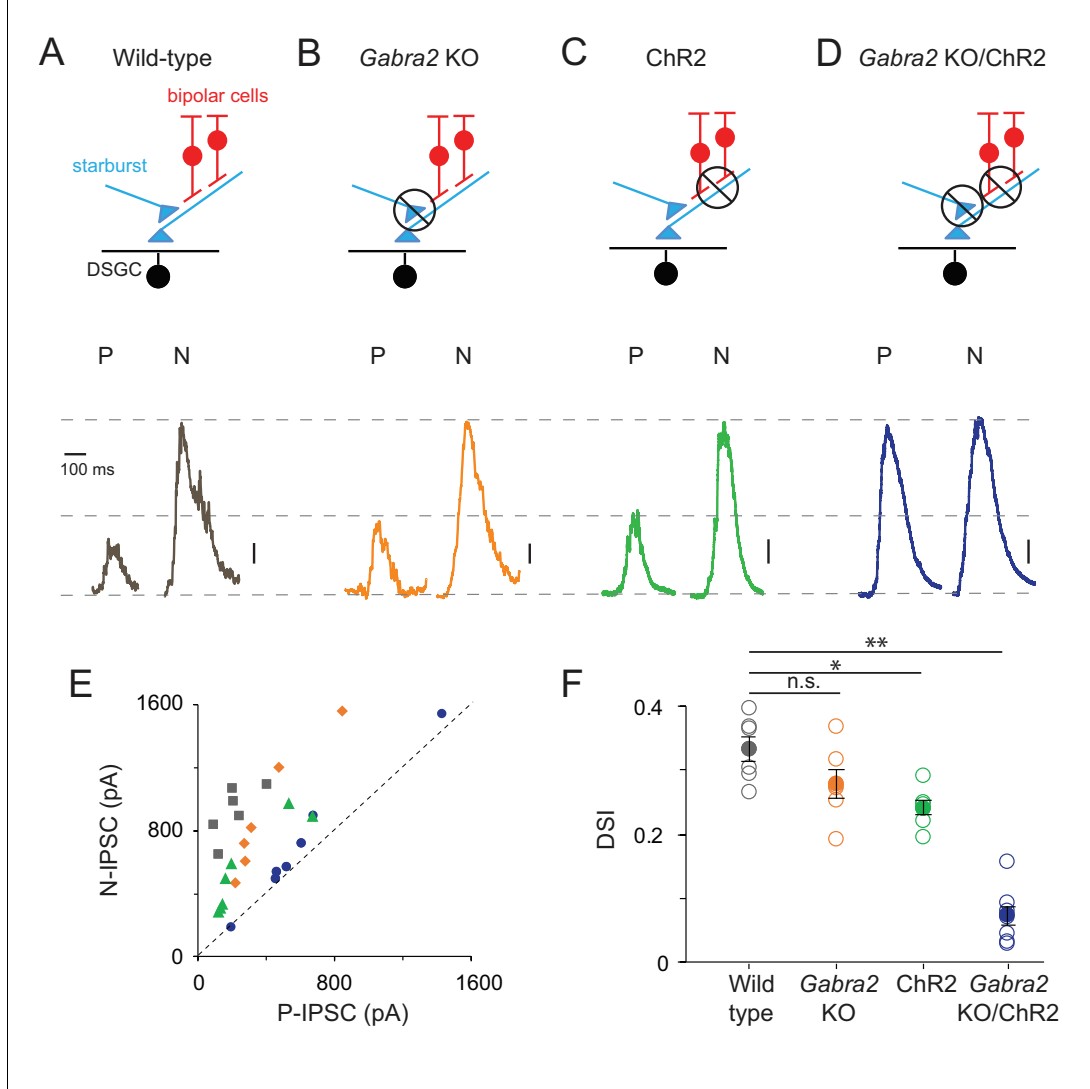

**Figure 2.** Direction selectivity in starburst dendrites relies on both excitatory and inhibitory network mechanisms (**A–D**) Starburst output monitored as IPSCs in voltage-clamped DSGCs (V$_{HOLD}$ ~0 mV) in a variety of mouse lines (as indicated). For simplicity, only ON IPSCs are shown for stimulus motion in the DSGC's preferred (P) or null-direction (N) under conditions in which (**A**) all synaptic inputs are intact; (**B**) mutual inhibition between starbursts is selectively disrupted using the conditional *Gabra2* KO; (**C**) bipolar cell inputs are blocked and ChR2-expressing starbursts are directly stimulated; (**D**), both mutual inhibition and bipolar cell inputs are blocked. Vertical scale bar = 100 pA (**A–C**) or 200 pA (**D**). Responses are averaged over three trials. (**E**) The peak amplitudes of the IPSCs evoked during preferred- and null-direction motion are plotted against each other for the conditions noted in (**A–D**). A reference line with slope = 1 (dashed line) is shown to facilitate comparisons. n = 6 for wild type, *Gabra2* KO and ChR2; n = 7 for *Gabra2* KO/ChR2. (**F**) The average direction-selectivity index (DSI) computed from the normalized vector sum of the peak amplitude of IPSCs evoked by stimuli moving in eight directions, for the conditions in (**A–D**) (See Materials and methods for DSI calculation; DSI = 0 indicates non-directional responses; DSI = 1 indicates strongly tuned responses). Pooled data are represented as mean ± SEM (Solid circles), while single cell responses are denoted by open circles. Statistical significance was estimated by unpaired t-tests, where *, ** or n.s. indicate p=0.0026, 0.0001 and 0.1068, respectively.
DOI: https://doi.org/10.7554/eLife.42392.003

The following source data is available for figure 2:

**Source data 1.** Included is an exccel file containing the data presented in *Figure 2A-F*.
DOI: https://doi.org/10.7554/eLife.42392.004

measured prior to the voltage-clamp experiments (*Figure 4B–D*). These results provided strong evidence that temporal asymmetries in E/I onsets alone can drive direction selectivity in ganglion cells. Moreover, since temporal asymmetries are observed under conditions in which photoreceptor/

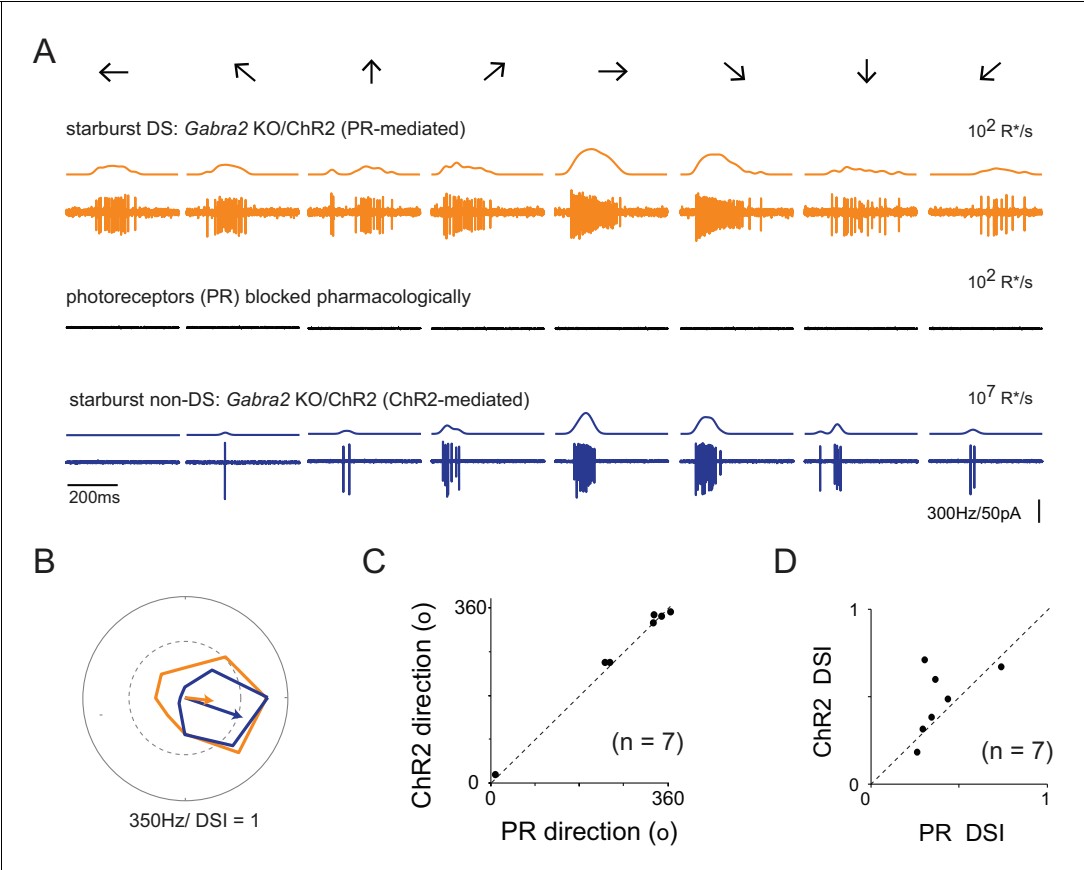

**Figure 3.** Retinal direction selectivity in the absence of asymmetric starburst amacrine cell responses  (A) Spiking responses from a DSGC in the *Gabra2* KO/ChR2 mouse line measured under conditions when photoreceptor (PR) output is intact (*top*), when photoreceptor synapses are blocked pharmacologically (50 µM DL-AP4, 10 µM UBP310 and 20 µM CNQX (*middle*)), and subsequently, when stimulus intensity is increased to directly activate starbursts using ChR2 (*bottom*). This enables a direct comparison of direction selectivity in a DSGC under conditions in which starburst output is DS (*top*) or non-DS (*bottom*). Stimuli were moved in eight directions indicated by the arrows at a velocity of 1200 µm/s. The smooth traces on top of each spike train indicates the average spike rate estimated by low-pass filtering the spike train via convolution with a Gaussian kernel (σ = 25 ms). (B) Polar plots of the peak spike rates for the responses in the two conditions shown in (A). The arrow indicates the DSGC's preferred direction, scaled to the DSI. (C–D) A comparison of the preferred direction (C) and DSI (D) of the responses evoked during ChR2 stimulation, and during intact photoreceptor stimulation in the *Gabra2* KO/ChR2 mouse line (n = 7).
DOI: https://doi.org/10.7554/eLife.42392.005

The following source data is available for figure 3:

**Source data 1.** Included is an exccel file containing the data presented in *Figure 3A-D*.
DOI: https://doi.org/10.7554/eLife.42392.006

glutamate receptors are blocked, it indicates that they arise from the starburst network itself, through a differential functional connectivity of cholinergic/GABAergic synapses to DSGCs (*Figure 1B,D*).

## Differential transmission of ACh/GABA under physiological conditions

Temporal asymmetries were also observed under natural conditions, i.e. when responses were driven by photoreceptors (*Figure 4E–H*), consistent with previous studies (*Fried et al., 2002*; *Kostadinov and Sanes, 2015*; *Pei et al., 2015*; *Sethuramanujam et al., 2016*; *Taylor and Vaney, 2002*). Similar to the optogenetic experiments, the magnitude of the E/I temporal offsets measured under physiological conditions (i.e. photoreceptor mediated) was greatest for preferred-direction motion and progressively decreased as the stimulus direction approached the DSGC's null-direction (*Figure 4F*). In these experiments, superior coding DSGCs were excluded from the analysis, since in these ganglion cells gap junctions are known to drive early excitatory inputs (*Trenholm et al., 2013*).

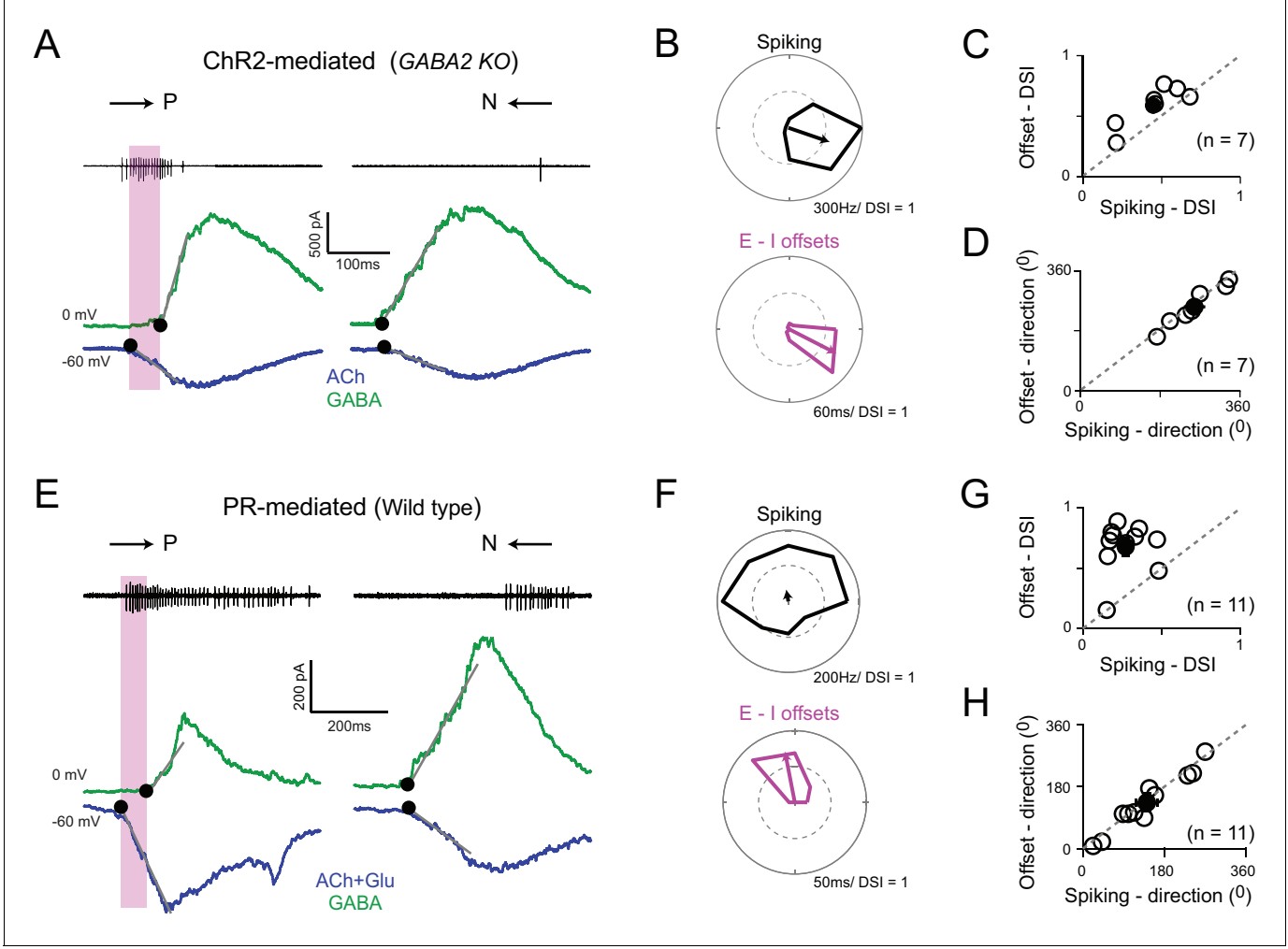

**Figure 4.** Directionally tuned E/I temporal offsets maintain direction selectivity in DSGCs in the absence of asymmetric starburst responses (**A**) Optogenetically evoked spikes, EPSCs and IPSCs recorded in succession from the same DSGC in the *Gabra2* KO/ChR2 mouse line (in the absence of photoreceptor input). The onset of excitation and inhibition (indicated by •), was estimated by back extrapolating the lines of best fit of the initial response (grey lines; see Materials and methods). The shaded box indicates the E/I temporal offset window during which a large fraction of the spikes appear to be generated. (**B**) Polar plots of the peak firing rates (*top*) and E/I temporal offsets (*bottom*) for optogenetic responses (depicted in **A**) to eight stimulus directions. **C**–**D**, A comparison of the DSI (**C**) and direction coded (**D**) estimated from the peak firing rates or E/I temporal offsets (as shown in **B**) across 7 cells (open circles). Pooled data (closed circles) are represented as mean ± SEM. (**E**–**H**) Similar to **A**–**D** but for responses measured under physiological conditions i.e. responses driven by photoreceptors in the wild-type retina (n = 11). The DSI of the E/I temporal offsets was higher than the DSI of spiking (p<0.001). Pooled data are represented as mean ± SEM.

DOI: https://doi.org/10.7554/eLife.42392.007

The following source data is available for figure 4:

**Source data 1.** Included is an exccel file containing the data presented in *Figure 4A-D*.
DOI: https://doi.org/10.7554/eLife.42392.008
**Source data 2.** Included is an exccel file containing the data presented in *Figure 4E-H*.
DOI: https://doi.org/10.7554/eLife.42392.009

The E/I temporal offset tuning was significantly sharper when compared to the tuning of the spiking responses, as indicated by the DSI (*Figure 4G*). In a later section, we use modeling to show that this difference arises because the DS spiking responses are shaped by two distinct mechanisms with different tuning properties (Figure 7). By contrast, the predicted direction based on offsets were almost perfectly aligned with the direction encoded based on the spiking responses (*Figure 4H*).

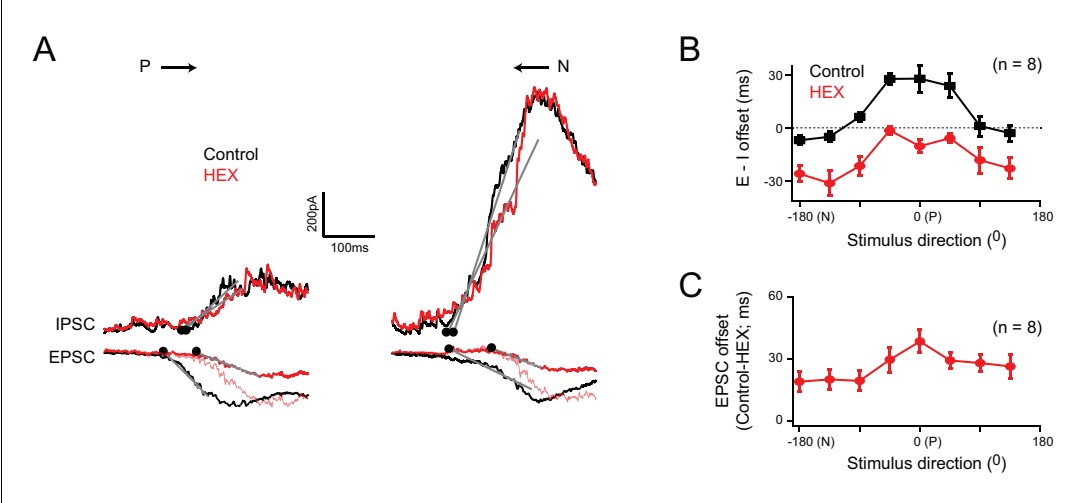

**Figure 5.** Cholinergic excitation shapes temporal E/I temporal offsets under physiological conditions (**A**) The EPSCs and IPSCs elicited in a DSGC for preferred and null stimuli, in control (black), and in the presence of a nicotinic acetylcholine receptor antagonist (hexamethonium, HEX; red traces). A normalized version of the EPSC measured in HEX (light red trace) is illustrated to emphasize the delay in excitation that occurs when cholinergic receptors are blocked. (**B**) The directional tuning of the E/I temporal offsets in control and HEX averaged across 8 cells. The offsets were aligned to the preferred direction (P) of the spike responses, which were measured in the same DSGC before performing the whole cell recording. HEX significantly reduced the E/I temporal offsets across all directions (p<0.01). Data are represented as mean ± SEM. (**C**) The directional tuning of the timing delay of the EPSC in HEX relative to control (n = 8). Data are represented as mean ± SEM. The increased onset latency observed in HEX continued to be observed when inhibition was blocked, indicating that they cannot be accounted for by large voltage–clamp errors associated with inhibitory conductances (*Figure 5—figure supplement 1*).
DOI: https://doi.org/10.7554/eLife.42392.014

The following source data and figure supplement are available for figure 5:

**Source data 1.** Included is an exccel file containing the data presented in *Figure 5*.
DOI: https://doi.org/10.7554/eLife.42392.016
**Figure supplement 1.** The EPSC timing delay in HEX is independent of inhibition.
DOI: https://doi.org/10.7554/eLife.42392.015

Consistent with the notion that E/I temporal offsets arise from a differential transmission of ACh/ GABA (*Figure 1D*), blocking nicotinic ACh (nACh) receptors using a specific antagonist (100 μM hexamethonium) increased the onset latency for excitation evoked by motion in the preferred-direction, and largely abolished the E/I temporal offsets in this direction (*Figure 5A*). For null-direction motion, blocking these receptors resulted in a negative offset, as the GABAergic inhibition arising from null-side starbursts was no longer balanced by cholinergic inputs but continued to arrive before glutamate inputs (*Figure 5A*). Further blocking GABAergic inhibition did not alter the delay induced by blocking nACh receptors, indicating that the E/I temporal offset estimates were not confounded by voltage-clamp errors associated with large inhibitory conductances (*Figure 5—figure supplement 1*; *Poleg-Polsky and Diamond, 2011*). Plotting the E/I temporal offsets for all stimulus directions reveals that hexamethonium uniformly reduces the E/I temporal offsets by ~25 ms (average delay = 26 ± 2 ms; n = 8) regardless of direction (*Figure 5B,C*), consistent with the idea that starbursts make cholinergic connections from all around the DSGC (*Figure 1D*). Given the stimulus moved at 1000 μm/s, this corresponds to a spatial offset of ~25 μm (see Discussion). It is also worth noting that, both ACh and GABA signals arising from starburst dendrites are spatially offset relative to glutamate inputs (*Figure 1B,D*), unlike the co-extensive glutamatergic/cholinergic receptive fields observed in rabbit retina (*Brombas et al., 2017*; *Fried et al., 2005*; *Lee et al., 2010*). Therefore, we conclude that the differential transmission of ACh/GABA produces E/I that are temporally synchronized in the null- but not preferred-direction (*Figure 1D* and *Figure 5A*), providing the substrate for a timing based mechanism for generating direction selectivity in ganglion cells.

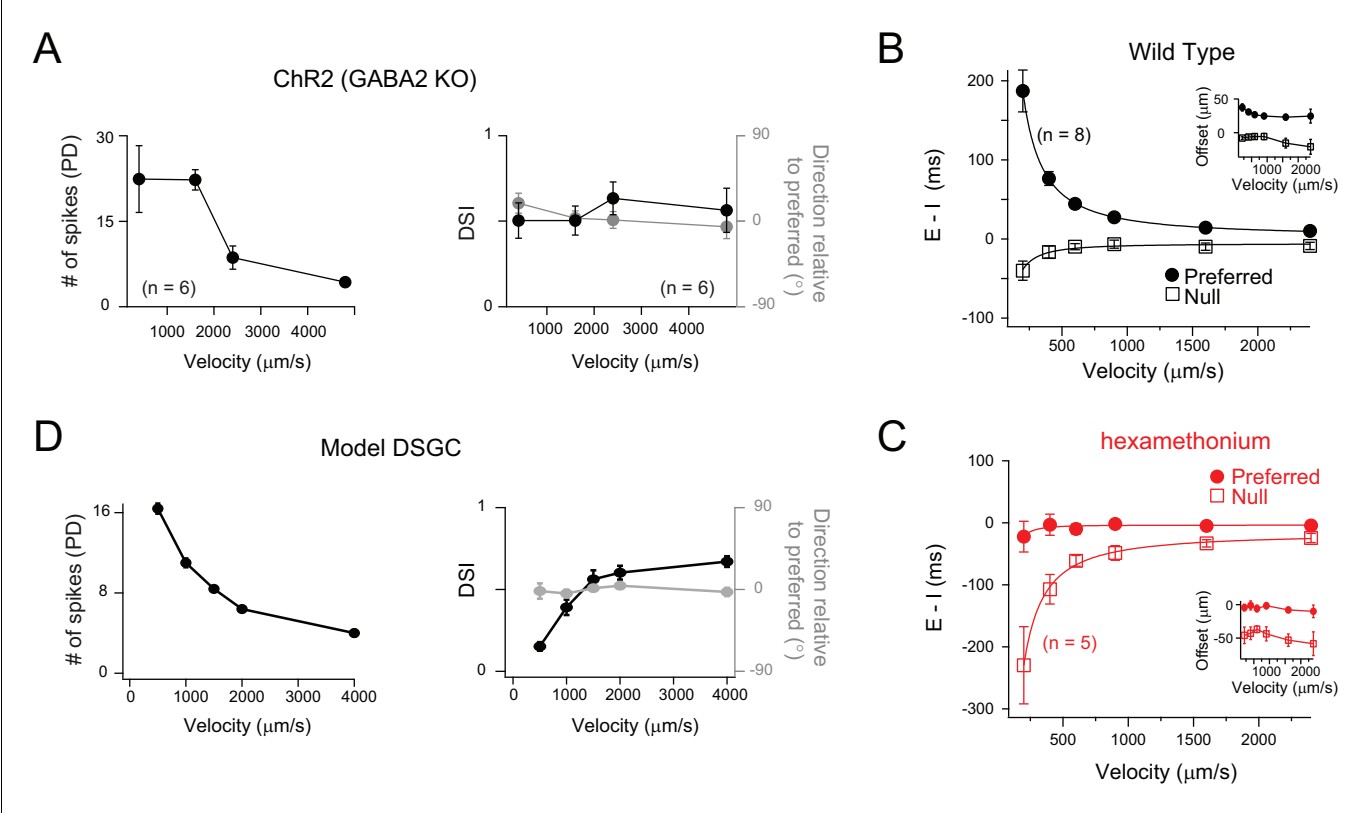

**Figure 6.** Velocity tuning of E/I temporal offsets in DSGCs. (**A**) A plot of the average number of ChR2-evoked spikes for preferred direction motion as a function of stimulus velocity (*left panel*; n = 6), measured from DSGCs in the *Gabra2* KO/ChR2 mouse line. The average DSI and preferred direction remains stable across velocity (*right panel*; n = 6). Data are represented as mean ± SEM. (**B–C**) The E/I temporal offsets measured from DSGCs in the wild-type retina, for both preferred and null directions as a function of velocity (**B**); n = 8. The offsets during preferred direction motion were significantly larger than those for null direction motion for all velocities (p<0.001). The temporal offsets correspond to a relatively fixed spatial offset (calculated by velocity x E/I temporal offsets) across a range of velocities tested (*inset*). The effects of blocking cholinergic transmission (hexamethonium) on E/I temporal offsets is shown in C). Data are represented as mean ± SEM. (**D**) Spike output of a computer simulated model DSGC in which the E/I ratio was fixed (1/2) but the E/I temporal offsets were varied according to measurements in B). E/I temporal offsets alone could generate robust DS across a range of velocities, except of the lowest velocity tested (see text for details).

DOI: https://doi.org/10.7554/eLife.42392.017

The following source data is available for figure 6:

**Source data 1.** Included is an exccel file containing the data presented in *Figure 6B-C*.

DOI: https://doi.org/10.7554/eLife.42392.018

## E/I temporal offset mechanisms contribute to direction selectivity over a range of velocities

Next, we examined how E/I temporal offsets could contribute to direction selectivity across a range of stimulus velocities. In the *Gabra2* KO-ChR2, where DS in DSGCs relies solely on temporal E/I offsets, the number of optogenetically evoked spikes were found to decrease as a function of velocity, as expected (*Figure 6A*). However, despite the drop in the spike responses at higher stimulus velocities, the direction selectivity was not altered (*Figure 6A*). As it is not possible to isolate the effects of temporal offsets under physiological conditions, instead, we measured the velocity dependence of the E/I temporal offsets under physiological conditions (*Figure 6B*), and used these values to drive DS responses in a model DSGC, generated in the NEURON environment (*Figure 6D*; see Materials and methods for details).

The amplitude of E/I temporal offsets measured in the DSGC's preferred direction reduced with velocity, but were still measurable at the highest velocities tested (2400 µm/s; *Figure 6B*). The values of the E/I temporal offsets measured at different speeds corresponded to a ~ 30 µm spatial offset

between excitatory and inhibitory receptive fields (*Figure 6B* inset). Addition of nACh receptor antagonists (hexamethonium) abolished the E/I temporal offsets, confirming that the spatial offsets are mediated by starburst inputs at all velocities (*Figure 6C*). Conversely, for null-direction motion, E/I were co-activated through most of the velocity range, except for the lower velocities where inhibition tended to lead excitation (*Figure 6B*). For null-direction motion, the nACh receptor antagonists revealed negative offsets, as previously noted (*Figure 6C*).

Next, to examine the impact of E/I temporal offsets at different velocities we generated a model DSGC with excitatory and inhibitory inputs having timing offsets that varied with direction as measured experimentally, but with their amplitude ratio constant (E:I set at 1:2 for all directions; *Figure 4F*). As with our optogenetic study, for preferred motion, the number of spikes decreased as a function of velocity (*Figure 6D*), but remained robustly directional through most of velocity range. Only at the lowest speed, where the edges moving in the null direction spent considerable time on the excitatory receptive field after inhibition ceased, the model generated a few null-direction spikes, resulting in a decreased DSI (*Figure 6D*). In real DSGC this does not appear to be an issue, indicating that GABAergic inhibition must be slower and more powerful than that implemented in our model. On the other hand, the direction encoded remained largely unaltered with velocity. Thus, the simplified model captures many aspects of the DSGC response properties, and supports the idea that the E/I temporal offset mechanism contributes to direction selectivity in ganglion cells over a wide range of velocities.

## Complementary roles for E/I temporal offset and amplitude based DS mechanisms

Finally, we explored the possible computational benefits offered by the distinct DS mechanisms relying on temporal and amplitude E/I asymmetries. When the spiking responses were averaged over the entire trial, the direction selectivity generated with or without DS inhibition was similar, and thus the DS mechanisms appear redundant (*Figure 3*). However, when the DS tuning was examined on a finer time-scale (*Figure 7*), it showed distinct characteristics in the early and peak phases of the response, raising the possibility that the two mechanisms operate on distinct time-scales. While the direction encoded remained constant throughout the response, the tuning curve broadened as the response approached its peak, resulting in a marked decrease in the DSI (*Figure 7A–D*). Moreover, the initial estimate of direction was more variable compared to that made during the peak of the response (standard deviation of early and peak responses were $16 \pm 2^0$; and $8 \pm 2^0$, respectively; *Figure 7B,E,F*). This difference in variability may reflect the ability of distal vs. proximal dendritic sites to effectively transmit directional information to the soma via dendritic spikes (*Sivyer and Williams, 2013*). It is also important to note that the broadening of the tuning curve did not arise simply from a thresholding effect (i.e. an iceberg effect), as the last spikes at the tail end of the response were not sharply tuned (data not shown).

Given that early spikes appear to be driven largely by cholinergic excitation (*Figure 7A*), but not by glutamate inputs (*Figure 7—figure supplement 1*), we envisioned that E/I temporal offsets are likely to be more important in determining direction selectivity during the initial phase of the response, while the later phase would likely be dominated by E/I amplitude differences. It is important to note that both DS mechanisms rely on transmitter release from starbursts and thus operate on roughly the same spatial scales. Next, we tested these ideas in our model DS circuit, where we could easily control the timing and amplitude of E/I, independently (see Materials and methods for details).

Indeed, the characteristic properties of the DSGC's direction tuning were reproduced in a model DSGC driven with both temporal and amplitude asymmetries in E/I that arise from the specific arrangement of GABA, ACh and glutamate inputs (*Figure 7G*; *Figure 7—figure supplement 2A* illustrates the synaptic inputs/spiking response of the model DSGC). When starbursts were rendered non-DS in the model DSGC (*Figure 7G*; *Figure 7—figure supplement 2B*), temporal asymmetries alone were sufficient to generate DS responses with sharp tuning (*Figure 7G*), as observed in the ChR2-*Gabra2* KO mouse. When E/I temporal offsets were removed, modulating the peak amplitude of inhibition generated robust DS responses, but with wider tuning (*Figure 7G*; *Figure 7—figure supplement 2C*). However, in this model lacking offsets, the early responses were lost (*Figure 7G*). Thus, the DS mechanisms based on temporal and amplitude asymmetries appear to be complementary, each conferring distinct advantages: the former enables DSGCs to respond in a direction-

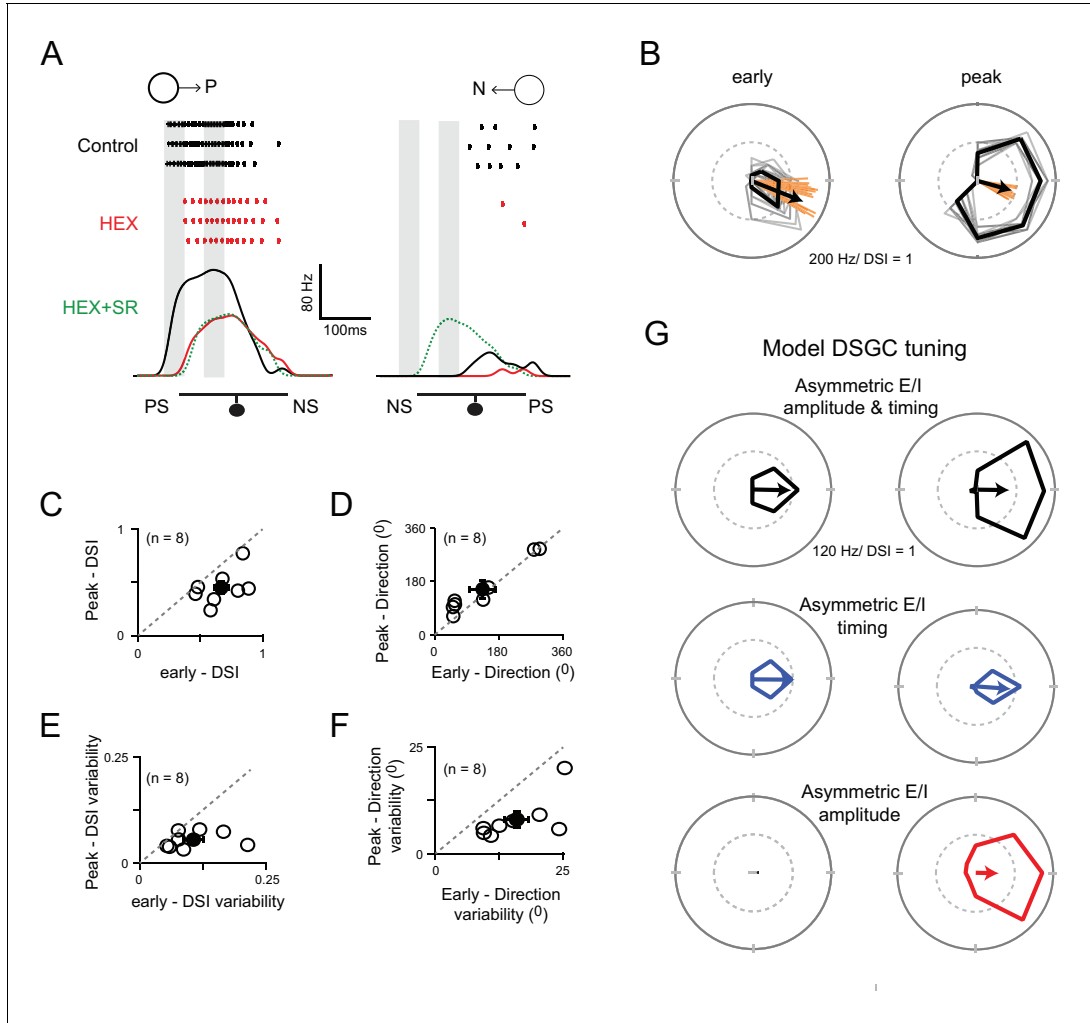

**Figure 7.** Two 'core' DS mechanisms are engaged during different phases of the DSGC response. (**A**) Spike rates during stimulus motion in the preferred or null directions in control conditions (black), after cholinergic excitation is blocked by HEX (red), and in the added presence of a GABA receptor blocker (5 µM SR-95531; green) to isolate the glutamate receptive field. Preferred and null responses could be accurately aligned, temporally, based on the glutamatergic receptive field measurement, which was important in estimating the temporal evolution of the DS tuning shown in (**B–G**). The effects of HEX on the response latency were reversible, and not observed when responses were weakened by blocking NMDA receptors (see *Figure 7—figure supplement 1*). PS and NS indicates the preferred- and null-sides of the DSGC. (**B**) Polar plots of a DSGC's early (*left*) and peak (*right*) spike responses (the 50 ms shaded regions in **A**) over 29 trials. The mean response is shown in black, while the individual trials are shown in grey. Note early responses are more sharply tuned but provide a less reliable indication of the direction (i.e. the direction encoded is more variable; orange lines) compared to the peak response. (**C–D**) A comparison of the average DSI (**C**) and preferred direction (**D**) calculated from the early and peak spike responses for individual cell (n = 8). The DSI of the early responses was significantly higher than the peak responses (p<0.01), while the direction encoded in both phases were near identical. Pooled data are represented as mean ± SEM. (**E–F**) A comparison of the trial-to-trial variability of the DSI (**E**) and preferred direction (**F**) of the early and peak responses (n = 8). Trial-to-trial variability was estimated by the standard deviation of DSI/preferred direction across trials within each cell (minimum 10 trials). The variability was higher for early responses for both DSI (p<0.05) and preferred direction (p<0.005). Pooled data are represented as mean ± SEM. (**G**) Tuning properties of a computer simulated model DSGC in which both E/I temporal offsets and amplitude asymmetries (*top*), only E/I temporal offsets (*middle*) or only E/I amplitude asymmetry (*bottom*) were implemented. E/I temporal offsets alone resulted in a sharper tuning similar to the early responses in B), but the tuning became broader when E/I ratios were intact, similar to the peak responses in B) (see *Figure 7—figure supplement 2* for synaptic currents and spiking responses measured in the model). Responses were averaged over 20 trials.

DOI: https://doi.org/10.7554/eLife.42392.010

The following source data and figure supplements are available for figure 7:

**Source data 1.** Included is an exccel file containing the data presented in *Figure 7*.

DOI: https://doi.org/10.7554/eLife.42392.013

**Figure supplement 1.** The effects of blocking nACh receptors on the latency of the DSGC response is reversible, and contrasts with the effects of NMDA receptor antagonists.

*Figure 7 continued on next page*

*Figure 7 continued*

DOI: https://doi.org/10.7554/eLife.42392.011

**Figure supplement 2.** Input/output measured in a model DSGC in which responses are driven by directional changes in inhibition and/or E/I temporal offsets

DOI: https://doi.org/10.7554/eLife.42392.012

selective manner sooner than they would have done otherwise, while the latter enables DSGCs to encode direction with higher fidelity albeit on a slower time scale.

## Discussion

### Parallel 'core' mechanisms generating direction selectivity in the retina

In the mammalian retina, direction selectivity is observed at the level of starburst dendrites as well as in downstream DSGCs. Our experiments using various pharmacological, optogenetic and cell-specific KO strategies to analyse synaptic inputs to DSGCs, suggest that multiple circuit mechanisms serve to generate direction-selectivity at both these points in the DS circuit.

At the level of starburst dendrites, three distinct mechanisms have been proposed to underlie direction selectivity. These include excitatory and inhibitory network mechanisms (*Ding et al., 2016*; *Fransen and Borghuis, 2017*; *Kim et al., 2014*; *Lee and Zhou, 2006*; *Münch and Werblin, 2006*), and cell intrinsic mechanisms (*Euler et al., 2002*; *Haussel et al., 2007*; *Tukker et al., 2004*). However, pinpointing the 'core' mechanism has been complicated because blocking these mechanisms individually, leaves starburst DS output relatively intact. Our result, using ChR2 to stimulate starburst output in a mouse line where inhibitory network mechanisms have been inactivated (*Chen et al., 2016*), demonstrate for the first time a condition where starburst release of GABA is non-directional. These results highlight the apparently redundant roles for excitatory and inhibitory network mechanisms and help amalgamate divergent views (*Ding et al., 2016*; *Fransen and Borghuis, 2017*; *Kim et al., 2014*; *Lee and Zhou, 2006*; *Münch and Werblin, 2006*).

While intrinsic DS mechanisms in starburst dendrites may be important for DS computations, the current results do not appear to reveal them in an obvious manner. This could be for several reasons. First, it is possible that the artificial activation of starbursts using optogenetics does not recruit dendritic voltage-gated channels that may underlie direction selectivity (reviewed in *Taylor and Smith, 2012*). Second, the removal of GABAergic inhibition between starbursts may hinder or occlude intrinsic DS mechanisms. Third, it is also possible that the limited set of stimuli (moving spots) which we utilized were not optimally designed to recruit cell intrinsic mechanisms. Thus, although DS is suppressed under conditions in which inhibitory and excitatory network mechanisms are blocked, we do not exclude the possibility that intrinsic mechanisms could serve as a third 'core' DS mechanism. A challenge for future experiments is to determine if and how each of these mechanisms work to maintain starburst DS output under different stimulus conditions.

At the level of DSGCs too, multiple DS mechanisms appear to be in play. Here, two fundamental mechanisms appear to originate from unique specializations of the starburst network. The first mechanism, which relies on amplitude asymmetries arising from the DS starburst dendrites (*Figure 1C*) and asymmetric starburst-DSGC wiring (*Figure 1B*), is well established (reviewed by *Mauss et al., 2017*; *Vaney et al., 2012*). The second mechanism relies on temporal asymmetries in excitation and inhibition (*Figure 1D*). While E/I temporal offsets have often been noted in the literature (*Fried et al., 2005*; *Kostadinov and Sanes, 2015*; *Pei et al., 2015*; *Sethuramanujam et al., 2016*; *Sivyer et al., 2010*; *Taylor and Vaney, 2002*), our results demonstrate for the first time that they are tuned precisely to the DSGC's preferred direction, and are alone sufficient to generate direction selective responses in ganglion cells.

Recent studies using voltage-clamp techniques have also shown the amplitude of the cholinergic signal to vary according to direction (*Lee et al., 2010*; *Pei et al., 2015*; *Park et al., 2014*; *Sethuramanujam et al., 2016*). However, these results need to be interpreted with caution. Dynamic changes in the inhibitory conductance could lead to serious voltage-clamp errors and estimates of the relative peak amplitude of cholinergic input in the preferred and null direction (*Poleg-Polsky and Diamond, 2011*; *Figure 5—figure supplement 1*). Thus, the development of alternate

methods, such as functional imaging, is required to fully resolve this issue. Importantly, the E/I offset mechanisms generating DS described here do not require amplitude changes in cholinergic inputs.

It is also worth noting that cholinergic, GABAergic and glutamatergic synapses are made relatively uniformly throughout the DSGC receptive field and are thought to constitute repeating dendritic DS subunits (*Barlow and Levick, 1965*; *Taylor et al., 2000*). Thus, although the temporal asymmetries are most apparent at response onset in our recordings made from the DSGC soma, they are likely present throughout the dendritic tree. Local E/I amplitude and timing differences are expected to shape dendritic responses, which may be further refined by dendritic non-linearities (*Oesch et al., 2005*; *Sivyer and Williams, 2013*; *Trenholm et al., 2014*). This is one reason why the spikes generated by E/I temporal offset mechanisms alone often outlast the E/I temporal offsets measured macroscopically (*Figure 7G*). The precise degree to which timing and amplitude differences in local inputs influence the DSGC's response directly depends on the extent to which dendrites operate independently (*Schachter et al., 2010*).

## The network of starburst amacrine cells produces temporal E/I asymmetries in DSGCs

Before the discovery of DS in starburst dendrites, classic models suggested that DS responses in ganglion cells were based on E/I timing differences (*Koch et al., 1982*; *Taylor et al., 2000*; *Torre and Poggio, 1978*). Previous studies have speculated that an intermediary cell in the inner and/or outer retina may control E/I timing differences at DSGCs (*Rivlin-Etzion et al., 2012*; *Taylor and Vaney, 2002*; *Vlasits et al., 2014*). In contrast, another study found that perturbing starburst cell morphology led to parallel changes in E/I amplitude ratio and E/I temporal offsets suggesting that these processes may be inextricably linked (*Kostadinov and Sanes, 2015*).

Our combined results from optogenetic, pharmacological and modelling experiments indicate that the temporal asymmetries in E/I are a natural consequence of the asymmetric/symmetric GABAergic/cholinergic connectivity of starburst cells to DSGCs (*Figure 1*; *Chen et al., 2016*; *Lee et al., 2010*; *Yonehara et al., 2011*). In this scheme, null-side starbursts extend the inhibitory receptive fields of DSGCs during null motion, producing a more powerful inhibition (*Fried et al., 2002*). Conversely, cholinergic excitation provided by preferred-side starbursts enable DSGCs to respond sooner to the moving edge than they would otherwise. We found that the degree to which cholinergic inputs advance the response is proportional to stimulus velocity, corresponding to a fixed spatial offset ~30 µm. In mouse retina, starbursts receive excitatory and inhibitory inputs on proximal dendrites but release transmitter from varicosities located on their distal dendrites (*Ding et al., 2016*; *Kim et al., 2014*; *Vlasits et al., 2016*). Mapping the receptive fields of individual starburst varicosities using $Ca^{2+}$ imaging shows them to be displaced by ~25 µm towards the soma (*Poleg-Polsky et al., 2018*). Thus, the extent to which cholinergic inputs expand the receptive field of the DSGC is well predicted by the input/output properties of starburst dendrites. The dimensions of the cholinergic receptive fields also indicate that despite being generated by paracrine mechanisms (which loosens the constraints set by the asymmetrical starburst-DSGC 'hard-wiring'; *Figure 1B*) (*Briggman et al., 2011*; *Brombas et al., 2017*), the distance over which ACh spreads appears relatively limited.

In contrast to our findings, in rabbit retina, cholinergic excitation is co-extensive with the glutamatergic receptive field and does not appear to extend the DSGC's receptive field beyond the dendritic field (*Brombas et al., 2017*; *Fried et al., 2005*; *Lee et al., 2010*; *Yang and Masland, 1992*). One reason for this could be that static stimuli employed in previous studies to measure the receptive field properties of DSGCs do not optimally stimulate starbursts, and therefore fail to reveal cholinergic excitation (*Fried et al., 2005*; *Yang and Masland, 1992*). Alternatively, in rabbit retina, the release of ACh may require a stronger stimulation of the starbursts (*Lee et al., 2010*). This causes a delay in the release of ACh, making it co-extensive with glutamate input (*Lee et al., 2010*). Indeed, under conditions where inhibition is blocked and starburst dendrites are more strongly stimulated, cholinergic inputs do provide lateral excitation (*Fried et al., 2005*; *Lee et al., 2010*; *Vaney et al., 2012*). Together, these findings have given rise to the notion that the role of ACh is modulatory (*Ariel and Daw, 1982*; *Amthor et al., 1996*; *Chiao and Masland, 2002*; *Lee et al., 2010*; but see *Pei et al., 2015*), in contrast to what we observed in mouse retina.

The demonstration that the parallel circuit mechanisms driving direction selectivity in ganglion cells are individually dispensable necessitates a re-evaluation of the conclusions drawn from a

multitude of studies carried out over the last several decades that considered a single 'core' DS mechanism. For example, numerous studies over the last forty years have found that blocking cholinergic receptors does not affect direction selectivity in DSGCs and have taken this to mean that ACh does not play an integral role in DS (reviewed by *Mauss et al., 2017*; *Vaney et al., 2012*), but rather provides an additional source of non-directional excitation to boost DSGC responses (*Brombas et al., 2017*; *Chen et al., 2016*; *Lee et al., 2010*), that is especially important under limiting conditions (*Sethuramanujam et al., 2016*; *Sethuramanujam et al., 2017*). In sharp contrast, our results here indicate that in fact ACh signals (unlike glutamate signals) are highly directional, not so much in their amplitudes as previously envisioned (*Chen et al., 2016*; *Grzywacz et al., 1998*; *Grzywacz et al., 1997*; *Lee et al., 2010*; *Pei et al., 2015*), but rather by virtue of their relative timing with inhibition. Cholinergic excitation and GABAergic inhibition are relatively imbalanced (or out of phase) during preferred-direction motion, but extremely well-balanced (or in phase) during null-direction motion, which is important for the computation (*Figure 1D*).

## Spatiotemporal dynamics of ACh, GABA and glutamate inputs

Traditional models of DS raised questions regarding the role of cholinergic excitation in driving direction selectivity. In the revised model (*Figure 1D*), the role of canonical bipolar cell glutamate pathways becomes questionable. Previously, we have shown that under low contrast conditions, glutamate appears to be mediated by NMDA receptors (*Sethuramanujam et al., 2016*; *Sethuramanujam et al., 2017*) and play an important role modulating cholinergic excitation. Under high contrast conditions, as we have used here, glutamate signaling is mediated by both AMPA and NMDA receptors. Glutamate input is relatively delayed and therefore makes the DSGC's preferred response more sustained. This contributes to changes of the DS tuning over time.

Moreover, since AMPA inputs are non-directional they are expected to broaden the DS tuning curve, by changing the E/I ratio across direction (*Poleg-Polsky and Diamond, 2016*). Indeed, a measurable broadening of the real DSGC tuning curves is noted at higher contrasts, although these effects are relatively small (*Sethuramanujam et al., 2017*). Additionally, given the difference in receptive fields between bipolar and starburst inputs to DSGCs, the E/I ratio is predicted to change over the duration of directional light responses (*Figure 1D*). Our model indicates that this effect is most pronounced during slow stimulus velocities, as temporal offsets caused by spatial differences are exacerbated as more time is spent between activations of each input to the DSGC, leading to a deterioration of DS (*Figure 6F*). However, in real DSGCs, DS is stable across velocities (*Brombas et al., 2017*), indicating that there must be additional mechanisms that compensate for these effects (e.g. inhibition from non-DS sources).

## Conclusions

Here we have shown how E/I amplitude and timing differences are orchestrated by parallel circuit mechanisms. In the case of pinpointing the mechanisms generating direction selectivity in starbursts (*Ding et al., 2016*; *Fransen and Borghuis, 2017*; *Kim et al., 2014*; *Lee and Zhou, 2006*; *Münch and Werblin, 2006*), or understanding whether timing (*Koch et al., 1982*; *Taylor et al., 2000*; *Torre and Poggio, 1978*) or amplitude ratios generate DS in ganglion cells, the realization of multiple DS mechanisms helps settle divergent views. In contrast, the suggestion of a central role for starburst ACh in driving E/I temporal offsets, necessitates re-enquiry into previous results which indicated a modulatory role.

Given the multiple DS mechanisms in the retina, it is interesting to speculate that DS neurons in higher visual areas are also created using a variety of circuit mechanisms. This may help explain conflicting results from recent studies regarding the origin of DS in cortex. For example, thalamocortical inputs to the visual cortex exhibit strong DS properties (*Sun et al., 2016*), apparently with retinal origin (*Cruz-Martín et al., 2014*). Yet, abolishing retinal direction selectivity has modest effects on cortical DS (*Hillier et al., 2017*), indicating that direction selectivity must also arise de novo through alternate parallel mechanisms (*Lien and Scanziani, 2018*; *Saul et al., 2005*; *Wilson et al., 2018*). Strikingly, in primate visual cortex, distinct mechanisms appear to shape early and late response (*Pack and Born, 2001*; *Thiele et al., 2004*) similar to what we have observed here for DSGCs, suggestive of a common organizational principle. It is also interesting to note that direction selectivity in other sensory areas including the auditory and somatosensory systems (*Kuo and Wu, 2012*;

*Wilent and Contreras, 2005*) may also rely on temporal and amplitude E/I asymmetries (*Li et al., 2015*; *Ye et al., 2010*; *Zhang et al., 2003*). While large-scale ultrastructural circuit mapping is bound to reveal circuit elements involved in the computation, it is only through the marriage of anatomical and physiological approaches can the circuit mechanisms be fully realized, as we have demonstrated here for the generation of direction selectivity at multiple levels in the mouse retina.

## Materials and methods

### Animals

Experiments were performed using adult (either sex) Trhr-EGFP (RRID: MMRRC_030036-UCD) or ChAT-IRES-Cre (RRID: MGI_5475195) crossed with Ai32 (RRID: MGI_5013789) with or without Gabra2$^{tm2.2Uru}$ (RRID: MGI_5140553). All procedures were performed in accordance with the Canadian Council on Animal Care and approved by the University of Victoria's Animal Care Committee.

### Physiological recordings

Mice were dark-adapted for approximately 30–60 min before being briefly anesthetized and decapitated. The retina was extracted and dissected in Ringer's solution under infrared light. The isolated retina was then mounted on a 0.22 mm membrane filter (Millipore) with a pre-cut window to allow light to reach the retina, enabling the preparation to be viewed with infrared light using a Spot RT3 CCD camera (Diagnostic Instruments) attached to an upright Olympus BX51 WI fluorescent microscope outfitted with a 40 × water immersion lens (Olympus Canada). The isolated retina was then perfused with warmed Ringer's solution (35–37°C) containing 110 mM NaCl, 2.5 mM KCl, 1 mM CaCl$_2$, 1. 6 mM MgCl$_2$, 10 mM dextrose and 22 mM NaHCO$_3$ that was bubbled with carbogen (95% O$_2$:5% CO$_2$).

DSGCs were identified by their genetic labeling or by their characteristic DS responses. Light stimuli, produced using a digital light projector (Hitachi Cpx1, refresh rate 75 Hz), were focused onto the outer segments of the photoreceptors using the sub-stage condenser. The background luminance, measured with a calibrated spectrophotometer (Ocean Optics), was set to ~10 photoisomerisations/s (R*/sec). Visual stimuli created in the Matlab environment (Psychtoolbox) were of positive contrasts, ranging between 15% and 1,000% (Weber contrast). Stimulus intensity was increased by five log units using neutral density filters to stimulate starburst-ChR2. Light-evoked activity was measured for 200 µm spot moving in eight directions at 1–1.6 mm/s.

Spike recordings were made with the loose cell-attached patch-clamp technique using 5–10 MΩ electrodes filled with Ringer's solution. Voltage-clamp whole-cell recordings were made using 4–7 MΩ electrodes containing 112. 5 mM CH3CsO3S, 7. 75 mM CsCl, 1 mM MgSO4, 10 mM EGTA, 10 mM HEPES, 5 mM QX-314 (Tocris) and 100 µM spermine (Abcam Biochemicals). The pH was adjusted to 7.4 with CsOH. The reversal potential for chloride was calculated to be –56 mV. The recordings were not corrected for junction potential. Recordings were made with a MultiClamp 700B amplifier (Molecular Devices). Signals were digitized at 10 kHz (PCI-6036E acquisition board, National 9 Instruments) and acquired using custom software written in LabVIEW developed by Dr. David Balya (Friedrich Miescher Institute, Switzerland; https://github.com/GBAlab/Presentinator). Unless otherwise noted, all reagents were purchased from Sigma-Aldrich Canada. D-AP5, and UBP310 were purchased from ABCAM Biochemicals. DL-AP4, SR-95531 and CNQX were purchased from Tocris.

### Computational modeling

Here we modified a detailed multi-compartmental model that was built previously by *Poleg-Polsky and Diamond, 2016* in the NEURON simulation environment (*Hines and Carnevale, 1997*) (https://github.com/geoffder/Spatial-Offset-DSGC-NEURON-Model). This model was based on a morphological reconstruction of a real DSGC (*Poleg-Polsky and Diamond, 2016*), with membrane capacitance and axial resistance set to 1 µF/cm2 and 100 Ωcm respectively. Membrane channels and noise were modelled using a stochastic Hodgkin and Huxley distributed mechanism (NEURON MOD file) as previously described. Non-voltage-gated leak conductances were set to reverse at −60 mV. Active membrane conductances were placed at the soma, primary dendrites and dendrites with the following densities (mS/cm$^2$) for soma, primary and terminal dendrites respectively: sodium (150/

150/30), potassium rectifier (70/70/35), delayed rectifier (3/0.8/0.4). Sodium and potassium conductances were blocked for voltage-clamp recordings. This enabled the model DSGC to process inputs actively i.e. via dendritic spikes (*Sivyer and Williams, 2013*).

In our model DSGC, synapses were placed on terminal dendrites throughout the dendritic arbor. Synapses housed glutamatergic (AMPA receptor), cholinergic (nicotinic receptor) and GABAergic (GABA$_A$ receptor) synaptic inputs, whose kinetic properties were based on experimentally measured miniature events. Synaptic inputs were activated by a simulated bar moving over the model DSGC in 8 directions at 1 mm/s. Cholinergic inputs were symmetrical across the simulated directions, with their probabilities of release (Pr) set at 0.5. Cholinergic inputs were also spatially offset by ~50 µm (corresponding to an E/I temporal offset of 50 ms for preferred stimuli moving at 1 mm/s), and for simplicity modeled based on fast modes of transmission, (similar to AMPA) rather paracrine transmission (see Discussion). AMPA receptor-mediated inputs were modelled to occur without any offset relative to their position on the DSGC's dendritic arbor. Thus they were simply activated on average (Pr ~0.5) when the simulated light stimulus passed over their synapse location. By contrast, GABAergic inhibitory inputs were made highly asymmetric, their Pr scaling down from ~0.5 in the null to ~0.012 in the preferred direction to simulate control conditions; or Pr was set constant at 0.5 to simulate conditions in which the starburst dendrites were non-DS. Their spatial offset ranged from ~50 µm in the null to ~0 µm in the preferred direction. The precise values across direction were estimated using sigmoidal fits to the tuning curves of the IPSCs (*Sethuramanujam et al., 2017*) and E/I temporal offsets measured experimentally (*Figure 4*):

$$\mathrm{Pr} = \mathrm{pPr} + (\mathrm{nPr} - \mathrm{pPr}) * (1.0 - 0.98/(1.0 + e^{(\mathrm{angle} - 91.0)/25.0)}))$$

$$\mathrm{spatial\,offset} = \mathrm{pOff} - (\mathrm{pOff} - \mathrm{nOff}) * (1.0 - 0.98/(1.0 + e^{(\mathrm{angle} - 74.69)/24.36)}))$$

Where, pPr, nPr refer to the probabilities of release in preferred and null directions, respectively. pOFF, nOFF refer to the spatial offsets in preferred and null directions, respectively; and angle refers to the direction of stimulated bar stimulus motion (0° is preferred, and 180° is null). Spatial offsets were then divided by the stimulus velocity to arrive at the final temporal offset. Importantly, the sequential activation of synapses by a moving bar stimulus, produced macroscopic inhibitory and excitatory currents which were well-matched to those measured experimentally using whole-cell patch-clamp techniques (*Figure 7—figure supplement 2*; *Figure 7*).

## Quantification and statistical analysis

DSI was calculated as (*Taylor and Vaney, 2002*):

$$DSI = \frac{\sum v_i}{\sum r_i}$$

where $v_i$ are vectors pointing in the direction of the stimulus and having length $r_i$, equal to the number of spikes recorded during that stimulus. DSI ranged from 0 to 1, with 0 indicating a perfectly symmetrical response, and 1 indicating a response in only one of eight directions. The angle was calculated from the direction of the resultant of $\Sigma v_i$.

The E/I temporal offset window was calculated with the following protocol. First, the 20–80% rise time of the synaptic currents (EPSCs or IPSCs) was fit by a straight line (*Figure 4*). The response onset latency was measured as the point at which the extrapolated linear fit crossed the x axis (time axis). The E/I temporal offsets were calculated as difference in the excitatory and inhibitory onsets. Positive offsets indicate that excitation leads inhibition and vice versa. For the purposes of estimating DSI, negative offsets were set to 0.

In order to determine the early and peak spike responses (*Figure 7*), the spike trains in control conditions were aligned to the edge of the glutamate receptive field, measured either by the spike activity or EPSCs in glutamate isolation i.e. in cholinergic and GABA receptor antagonists (*Figure 7A*). After alignment, the number of spikes occurring before the stimulus entered the glutamate receptive field (~50 ms in the preferred direction) was considered as the early phase responses. These spikes were completely blocked by cholinergic receptor antagonists (*Figure 7A*, *Figure 7—*

*figure supplement 1*). The peak phase was estimated as a ~ 50 ms region close to the peak firing rate in the preferred direction (*Figure 7A*).

Population data have been expressed as mean ± SEM and are indicated in the figure legend along with the number of samples. Student's *t* test was used to compare values under different conditions and the differences were considered significant when p≤0.05, unless otherwise noted in the figure legend.

---

## Additional information

### Funding

| Funder | Grant reference number | Author |
|---|---|---|
| Canadian Institutes of Health Research | 159444 | Gautam B Awatramani |

The funders had no role in study design, data collection and interpretation, or the decision to submit the work for publication.

### Author contributions

Laura Hanson, Data curation, Formal analysis, Investigation, Figures 2, 3, 4A-D, 6A; Santhosh Sethuramanujam, Data curation, Writing—review and editing, All experiments in Figures 4E-F, 5,6B-C & 7A-F; Geoff deRosenroll, Data curation, Writing—review and editing, All modeling in Figures 6 and 7; Varsha Jain, Methodology, Figure 5—figure supplement 1; Gautam B Awatramani, Conceptualization, Supervision, Funding acquisition, Writing—original draft, Project administration, Writing—review and editing

### Author ORCIDs

Laura Hanson (iD) https://orcid.org/0000-0002-8737-9513
Santhosh Sethuramanujam (iD) http://orcid.org/0000-0001-6199-1221
Geoff deRosenroll (iD) http://orcid.org/0000-0002-5431-2814
Varsha Jain (iD) http://orcid.org/0000-0002-1620-4177
Gautam B Awatramani (iD) http://orcid.org/0000-0002-0610-5271

### Ethics

Animal experimentation: All procedures were performed in accordance with the Canadian Council on Animal Care and approved by the Animal Care Committee (protocol 2016-015) of the University of Victoria

### Decision letter and Author response

Decision letter https://doi.org/10.7554/eLife.42392.021
Author response https://doi.org/10.7554/eLife.42392.022

---

## Additional files

### Supplementary files

• Transparent reporting form
DOI: https://doi.org/10.7554/eLife.42392.019

### Data availability

All data generated or analysed during this study are included in the manuscript and supporting files.

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
