## [Decision Letter]

Thank you for submitting your article "Retinal direction selectivity in the absence of asymmetric starburst amacrine cell responses" for consideration by *eLife*. Your article has been reviewed by three peer reviewers, including Fred Rieke as the Reviewing Editor and Reviewer #1, and the evaluation has been overseen by Ronald Calabrese as the Senior Editor.

The reviewers have discussed the reviews with one another and the Reviewing Editor has drafted this decision to help you prepare a revised submission.

The reviewers were all excited about the potential of this work to transform how we think about coding of direction in the retina. Several substantive points arose in the reviews, and these were reinforced in consultation among the reviewers. The most salient of those are briefly described below, with more detail in the individual reviews.

1) Relation between space and time. How a spatial offset gets converted to a difference in timing deserves more attention. In particular, whether (and if so how) such a mechanism can account for the speed invariance of directional tuning should be explored. This conversion is central to the proposed mechanism, and hence clearly establishing and describing it is critical to the paper.

2) Presentation. The presentation of the work needs to be improved in several ways. This includes some specific figures, and accessibility to a non-specialist. See the reviews below for specifics and suggestions.

*Reviewer #1:*

This paper provides new insights into the computations underlying directionally-selective responses of retinal ganglion cells. A compelling explanation for directional selectivity has emerged from work from several laboratories over the past 15 years. This picture is based on spatial asymmetries in excitatory and inhibitory inputs to DSGCs, and the recruitment of stronger inhibitory input for null direction compared to preferred direction motion. The current paper suggests that this picture is incomplete, and that a second mechanism also contributes to directional selectivity. The new proposed mechanism is based on differences in the timing of excitatory and inhibitory input in preferred and null directions. The paper is clearly written and the conclusions generally are well supported. There are several issues, however, that I think need to be strengthened:

1) Timing of excitatory and inhibitory input (Figure 3). The similarity of the relative timing of excitatory and inhibitory input in the absence of ACh signaling and the corresponding slowing of excitatory input are important parts of the paper. Yet the data for these points is not presented in a particularly convincing way. First, it would be helpful to see the excitatory inputs in the same cell before and after applying HEX. Second, the relative timing of excitatory and inhibitory inputs is difficult to appreciate given the differences in scale (particularly true in Figure 3G, but also to a lesser extent in Figure 3A and D). Can you add a normalized version of those traces to facilitate a temporal comparison?

2) Variability in magnitude of inhibitory input. The measured inhibitory inputs vary widely in amplitude (e.g. Figure 1E) – even when recorded for similar stimuli. This raises concerns about experiments in which comparisons are made across recording conditions or across mice. Do you have an explanation for the variability in inhibitory input? Some discussion of that would be helpful.

3) Ch2R activation. It is not clear from the main text how Ch2R activation was used to produce motion signals in the SACs. This is slightly clearer from reading the methods, but only filling in some retina-specific details about the orderly spatial arrangement of SACs. This should get described clearly in the Results with a non-retina reader in mind.

4) Details of modeling. The model of Figure 4 needs to be described in considerably more detail. One specific issue is that it is not clear how much of the model is constrained from experiment. A second issue is how spikes are generated in the model.

*Reviewer #2:*

The manuscript by Hanson et al. reports a major refinement of the well-studied model of the direction-selectivity (DS) circuit in the mammalian retina. The key insight is that the temporal asymmetry between excitation (both ACh and gluamate-driven) and inhibition *alone* can drive DS in RGCs, even without a contribution from the DS computation of the starburst dendrites, which causes null direction inhibition to be larger than that in the preferred direction.

Despite a huge amount of previous work on this circuit and hints of this phenomenon in previous work (which the authors cite), this is an important contribution. In general, the manuscript is well written, and the data are clear and compelling.

My major concern is that the authors need to do a better job connecting the spatial offset that underlies this phenomenon to its description as a temporal offset. The experiments presented in Figure 2B explore a large range of speeds and show that the temporal offset DS computation is robust. How is this the case? This deserves more exploration. If the E vs. I RFs are spatially offset, as suggested, and that is the whole story, the mechanism should only work over a reasonably small range of speeds. Do some speeds depend on the ACh to GABA offset and others depend on the glutamate to GABA offset? Since the ACh RF is larger, it should be tuned to higher speeds.

A nice experiment to clear this up would be a careful measurement of the RF of each of the 3 synaptic components (ACh, glutamate, and GABA) with small spots of light. The authors have the genetic and pharamacological tools to make these measurements. With these spatial RFs, one could construct a simple linear model of how they could give you a temporal offset and over what speeds the temporal offset would be effective in preventing null direction spikes.

*Reviewer #3:*

The paper by Hanson and colleagues investigates, and answers, two central questions concerning direction selectivity in the mammalian retina: a) what is the mechanism underlying direction selectivity in starburst amacrine cells?, and b) what is the mechanism underlying direction selectivity in the direction selective ganglion cells? In brief, knowledge about the first question allows the authors to make starburst amacrine cells non-directional and to investigate the source of direction selectivity in ganglion cells. Surprisingly, they find ganglion cells to remain directionally selective when fed with non-directional signals from amacrine cells. As demonstrated by further analysis, the relative timing of excitatory and inhibitory input from amacrine cells is sufficient for direction selectivity, even when the amplitudes of these signals are the same for preferred and null direction motion.

To me, this is an extremely important piece of work, which will have a strong impact on the field. My only concern is the way the data are presented: here I see room for improvement.

Quite in general, the two questions mentioned above need to be better separated and presented as such. For someone not in the field, the discovery about the origin of direction selectivity in amacrine cells is almost invisible. The authors should clearly mark their findings as such and discuss what kind of model for amacrine cell direction selectivity they favor and which can be firmly excluded. As an example, the way I understand the data, one can now exclude a pure electrotonic summation of postsynaptic excitation as a source for direction selectivity: without inhibitory amacrine-amacrine synapses, no direction selectivity remains in response to channelrhodopsin-stimulation. Along the same line, the authors need to discuss the source of the timing difference between the excitatory and the inhibitory inputs to ganglion cells and present a network model, at least a scheme, which explains their findings. Lastly, the role of cholinergic transmission is only explicitly mentioned at the end of the manuscript: I would put the mysterious role of acetylcholine in this circuit right in the beginning and get back to it at the end.

As another general comment, the manuscript in its current form is rather inaccessible to non-specialists: the circuit needs to be introduced and current hypotheses need to be explained. Only then can the reader fully appreciate the astonishing results. The current manuscript seems to be written as a short report, but there is no need for that: it should have full length with more figures.

Once the manuscript is structured and extended in the way outlined above, this should make a beautiful publication and a major contribution to our understanding of the mechanism underlying direction selectivity in the retina. Congratulations!

[Editors' note: further revisions were requested prior to acceptance, as described below.]

Thank you for submitting your article "Retinal direction selectivity in the absence of asymmetric starburst amacrine cell responses" for consideration by *eLife*. Your article has been re-reviewed by Fred Rieke as the Reviewing Editor, and the evaluation has been overseen by Ronald Calabrese as the Senior Editor.

This is a revision of a paper describing a new mechanism for direction selectivity in the retina. The authors did a good job responding to the first set of reviews, and the paper has improved considerably. I think there are a few additional changes that could help improve the clarity of the paper for a general audience, as detailed below.

Figure 1: This figure sets up the paper nicely and helps clarify the issues to be tested. I think this could be further improved in two ways. First, it would be nice to include a simple anatomical diagram in the figure showing connections between starburst amacrines and DS ganglion cells. This could include the paracrine nature of ACh release and the asymmetric GABAergic synapses. Second, there are several places in the text in which referring back to the figure could be helpful to remind a reader of the basic model.

Subsection “Retinal direction selectivity in the absence of asymmetric starburst amacrine cell responses”, second paragraph: the transition here is abrupt. It would also help to refer back to Figure 1B here.

Subsection “Differential transmission of ACh/GABA under physiological conditions”, second paragraph: the different spatial offsets that are important could get laid out more clearly.

Subsection “Complementary roles for E/I offset and amplitude based DS mechanisms”, second paragraph: this could get elaborated a bit as it is not immediately clear how the broadening of tuning coincides with an increase in reliability of estimated direction.

---

## [Author Response]

The reviewers were all excited about the potential of this work to transform how we think about coding of direction in the retina. Several substantive points arose in the reviews, and these were reinforced in consultation among the reviewers. The most salient of those are briefly described below, with more detail in the individual reviews.1) Relation between space and time. How a spatial offset gets converted to a difference in timing deserves more attention. In particular, whether (and if so how) such a mechanism can account for the speed invariance of directional tuning should be explored. This conversion is central to the proposed mechanism, and hence clearly establishing and describing it is critical to the paper.2) Presentation. The presentation of the work needs to be improved in several ways. This includes some specific figures, and accessibility to a non-specialist. See the reviews below for specifics and suggestions.

We thank the reviewers for their constructive comments and enthusiasm. We fully agree with the consensus critiques and have taken substantive steps to address the two major concerns outlined above.

First, to clarify relationship between space and time we have conducted several new experiments:

– We examined E/I offsets at different velocities

– We tested the role of cholinergic excitation in mediating offsets at different velocities using pharmacology

– We constructed a computer model DSGC, which was driven with a fixed E/I ratio, but variable timing offsets (according to our experimental results mentioned above).

Results from these new experiments and analyses help define the role of E/I offsets over a range of velocities and clarify how spatial offsets get converted into timing differences.

Second, to improve the general presentation we have thoroughly re-worked the manuscript. We now include:

– A simple schematic of the DS circuit mechanisms has been presented upfront to make the proposed hypotheses clear and accessible to the non-specialist.

– As suggested, we have expanded the paper from a short to a long format (4 figures to 7 figures)

– We now have emphasized both the major advances i.e. DS mechanisms in starburst dendrites as well as DS mechanisms at the level of DSGCs

– As suggested by the reviewer, we now discuss the role of acetylcholine in the DS computation sooner in the Introduction

– Figures have been modified to better display the measured E/I offsets and the effects of cholinergic blockers (see responses to specific queries below).

– A broader discussion of parallel mechanisms generating direction selectivity in visual cortex and other areas of brain has been added.

Overall, we feel these changes have strengthened the conclusions of this study, and once again thank all the reviewers for the insightful feedback.

Reviewer #1:[…] The paper is clearly written and the conclusions generally are well supported. There are several issues, however, that I think need to be strengthened:1) Timing of excitatory and inhibitory input (Figure 3). The similarity of the relative timing of excitatory and inhibitory input in the absence of ACh signaling and the corresponding slowing of excitatory input are important parts of the paper. Yet the data for these points is not presented in a particularly convincing way. First, it would be helpful to see the excitatory inputs in the same cell before and after applying HEX. Second, the relative timing of excitatory and inhibitory inputs is difficult to appreciate given the differences in scale (particularly true in Figure 3G, but also to a lesser extent in Figure 3A and D). Can you add a normalized version of those traces to facilitate a temporal comparison?

We have taken several steps to better characterize and more clearly present E/I timing differences, as described below:

– We have now described how onsets of excitation and inhibition were measured and marked the onsets on current traces, and shaded E/I windows (Figure 4)

– As suggested, we now show EPSCs/IPSCs recorded from the same DSGC, before and after application of HEX (now Figure 5)

– We also quantify the effect of HEX on E/I offsets across 8 stimulus directions, and demonstrate that excitation is consistently delayed by ~25ms (Figure 5)

– We performed new control experiments to ensure that delays observed in HEX do not arise from voltage-clamp errors associated with large inhibitory conductances (Figure 5—figure supplement 1).

– We also measured E/I offsets in the presence of HEX across a range of velocities (Figure 6B)

– To facilitate comparison of the EPSC timing differences in control and HEX, we have added the normalized versions of the EPSCs in Figure 5 and Figure 5—figure supplement 1.

2) Variability in magnitude of inhibitory input. The measured inhibitory inputs vary widely in amplitude (e.g. Figure 1E) – even when recorded for similar stimuli. This raises concerns about experiments in which comparisons are made across recording conditions or across mice. Do you have an explanation for the variability in inhibitory input? Some discussion of that would be helpful.

The variability in the IPSCs (now in Figure 2E) could arise from differences in ChR2 expression levels and/or run down of ChR2 responses over the course of the recording session (which could last up to 8 hours). However, within a single cell we found that the responses and the directional tuning was stable across trials and hence we chose to include the weaker responses in our dataset. These points have been included in the Results section.

3) Ch2R activation. It is not clear from the main text how Ch2R activation was used to produce motion signals in the SACs. This is slightly clearer from reading the methods, but only filling in some retina-specific details about the orderly spatial arrangement of SACs. This should get described clearly in the Results with a non-retina reader in mind.

In the revised manuscript, we better describe the DS circuit in the Introduction, highlighting the inhibitory connections between anti-parallel SAC dendrites.

We have also included details in the Results section regarding the methods used for ChR2 activation and how this technique was used to manipulate the starburst mechanisms of direction selectivity.

4) Details of modeling. The model of Figure 4 needs to be described in considerably more detail. One specific issue is that it is not clear how much of the model is constrained from experiment.

In the revised methods, the computational model has been presented in greater detail. In the model, the kinetics of the synaptic inputs were modeled based on experimentally measured miniature events, and as such produce macroscopic inhibitory and excitatory currents that are well-matched to those measured experimentally at the soma. These are illustrated in Figure 7—figure supplement 2. The code for the model has also been made publicly available (https://github.com/geoffder/Spatial-Offset-DSGC-NEURON-Model).

A second issue is how spikes are generated in the model.

In our model, spikes originate in the active dendritic arbours of the DSGC. Dendrites and the soma were endowed with voltage-gated sodium and potassium conductances. This enabled local synaptic inputs to generate dendritic spikes, which forward propagate and generate full-blown somatic action potentials with high fidelity, as observed experimentally (Sivyer et al., 2013). This information has been included in the Materials and methods section.

Reviewer #2:[…] In general, the manuscript is well written, and the data are clear and compelling.My major concern is that the authors need to do a better job connecting the spatial offset that underlies this phenomenon to its description as a temporal offset. The experiments presented in Figure 2B explore a large range of speeds and show that the temporal offset DS computation is robust. How is this the case? This deserves more exploration. If the E vs. I RFs are spatially offset, as suggested, and that is the whole story, the mechanism should only work over a reasonably small range of speeds. Do some speeds depend on the ACh to GABA offset and others depend on the glutamate to GABA offset? Since the ACh RF is larger, it should be tuned to higher speeds.

We took several steps to do a better job of relating spatial and temporal offsets:

– We examined E/I offsets at different velocities (new Figure 6B)

– We tested the role of cholinergic excitation in mediating offsets at different velocities using pharmacology (new Figure 6B)

– In the insets in Figure 6B, we plot the spatial offsets (estimated as velocity * temporal offset) and show it to be relatively constant across velocity

– We constructed a computer model DSGC, which was driven with a fixed E/I ratio but variable timing offsets (according to our experimental results mentioned above). As the reviewer suspects, the direction selectivity was stable for most velocities, except the slowest velocity tested (Figure 6). However, in real DSGCs, DSI was stable across velocity indicating additional mechanisms may exist to compensate for the expected deterioration of DS arising from spatial offsets. We have added these points to the Discussion section. (Also see the response to reviewer 1’s first question.)

– Also, while it is true that ACh RF is larger, it is dynamic. Only one half is activated for every direction, since the release of ACh is presumably directional.

A nice experiment to clear this up would be a careful measurement of the RF of each of the 3 synaptic components (ACh, glutamate, and GABA) with small spots of light. The authors have the genetic and pharamacological tools to make these measurements. With these spatial RFs, one could construct a simple linear model of how they could give you a temporal offset and over what speeds the temporal offset would be effective in preventing null direction spikes.

RF measurements using small spots might not reveal the spatial extent of cholinergic excitation accurately, as static stimuli do not optimally stimulate starburst dendrites. In fact, such RF measurements made in the rabbit DSGCs failed to reveal offset cholinergic inputs (Fried et al., 2005; Yang and Masland, 1994). Thus, we felt the simple mapping technique was not likely to be fruitful. We have included these points in the Discussion. On the other hand, our biophysical model using experimentally driven synaptic inputs (in response to moving stimuli) nicely recapitulates the DSGC’s response properties, including the temporal evolution of the DS tuning (Figure 7).

Reviewer #3:[…] To me, this is an extremely important piece of work, which will have a strong impact on the field. My only concern is the way the data are presented: here I see room for improvement.Quite in general, the two questions mentioned above need to be better separated and presented as such. For someone not in the field, the discovery about the origin of direction selectivity in amacrine cells is almost invisible. The authors should clearly mark their findings as such and discuss what kind of model for amacrine cell direction selectivity they favor and which can be firmly excluded. As an example, the way I understand the data, one can now exclude a pure electrotonic summation of postsynaptic excitation as a source for direction selectivity: without inhibitory amacrine-amacrine synapses, no direction selectivity remains in response to channelrhodopsin-stimulation.

In the revised manuscript we now make it clear in the Introduction that this study makes two major advancements; one relating to DS in amacrine cell dendrites, the other to DS in ganglion cells

The reviewer is correct in stating that dendritic filtering is unlikely to play an important role. However, we choose to interpret the data conservatively for the following reasons. First, it is not clear how ChR2 stimulates starburst release. It is possible that the artificial activation of starbursts using optogenetics does not recruit dendritic voltage-gated channels that may underlie direction selectivity. Second, the removal of GABAergic inhibition between starbursts may hinder or occlude intrinsic DS mechanisms. Third, it is also possible that the limited set of stimuli (moving spots) which we utilized were not optimally designed to recruit cell intrinsic mechanisms. These points are made in the Discussion section.

Along the same line, the authors need to discuss the source of the timing difference between the excitatory and the inhibitory inputs to ganglion cells and present a network model, at least a scheme, which explains their findings.

As mentioned previously, we now present schematics of the different possible configurations of the DS circuit and highlight how offset cholinergic excitation might generate E/I timing differences (Figure 1).

Lastly, the role of cholinergic transmission is only explicitly mentioned at the end of the manuscript: I would put the mysterious role of acetyl choline in this circuit right in the beginning and get back to it at the end.As another general comment, the manuscript in its current form is rather inaccessible to non-specialists: the circuit needs to be introduced and current hypotheses need to be explained. Only then can the reader fully appreciate the astonishing results. The current manuscript seems to be written as a short report, but there is no need for that: it should have full length with more figures.

Thanks for these suggestions. We have now introduced the role of ACh earlier and reformatted the manuscript to a full-length form with 7 figures (instead of the original 4). We hope in the revised form it is more accessible to the non-specialist.

[Editors' note: further revisions were requested prior to acceptance, as described below.]

This is a revision of a paper describing a new mechanism for direction selectivity in the retina. The authors did a good job responding to the first set of reviews, and the paper has improved considerably. I think there are a few additional changes that could help improve the clarity of the paper for a general audience, as detailed below.Figure 1: This figure sets up the paper nicely and helps clarify the issues to be tested. I think this could be further improved in two ways. First, it would be nice to include a simple anatomical diagram in the figure showing connections between starburst amacrines and DS ganglion cells. This could include the paracrine nature of ACh release and the asymmetric GABAergic synapses. Second, there are several places in the text in which referring back to the figure could be helpful to remind a reader of the basic model.

We have now added two new diagrams to illustrate the DSGC circuitry (Figure 1A) and the starburst-DSGC connectivity (Figure 1B), to better understand why paracrine ACh has been postulated. We have referred to these diagrams wherever appropriate.

Subsection “Retinal direction selectivity in the absence of asymmetric starburst amacrine cell responses”, second paragraph: the transition here is abrupt. It would also help to refer back to Figure 1B here.

We have added the following statement to smooth the transition: “We hypothesized that asymmetries in the timing of E and I arising from the differential functional starburst-DSGC connectivity gives rise to a parallel DS mechanism (Figure 1D).”

Subsection “Differential transmission of ACh/GABA under physiological conditions”, second paragraph: the different spatial offsets that are important could get laid out more clearly.

We have re-worked this section and moved it to the Discussion. It now reads:

“We found that the degree which cholinergic inputs advance the response is proportional to stimulus velocity, corresponding to a fixed spatial offset ~ 30 µm. […] Thus, the extent to which cholinergic inputs expand the receptive field of the DSGC is well predicted by the input/output properties of starburst dendrites.”

Subsection “Complementary roles for E/I offset and amplitude based DS mechanisms”, second paragraph: this could get elaborated a bit as it is not immediately clear how the broadening of tuning coincides with an increase in reliability of estimated direction.

We now offer a possible explanation for the changes in variability: “While the direction encoded remained constant throughout the response, the tuning curve broadened as the response approached its peak, resulting in a marked decrease in the DSI (Figure 7A-D). […] It is also important to note that the broadening of the tuning curve did not arise simply from a thresholding effect (i.e. an iceberg effect), as the last spikes at the tail end of the response were not sharply tuned (data not shown).”